# Stablecoin-Based Digital Trading and Investment Platforms and Their Potential in Overcoming Sanctions Restrictions

**Elena Vladimirovna Travkina ***, **Alim Borisovich Fiapshev ***  and **Marianna Tolevna Belova ***

Department of Banking and Monetary Regulation, Financial University under the Government of the Russian Federation, 125167 Moscow, Russia
* Correspondence: evtravkina@fa.ru (E.V.T.); abfiapshev@fa.ru (A.B.F.); mtbelova@fa.ru (M.T.B.)

**Abstract:** The current article summarizes the main properties of stablecoins and explores their potential use in digital platforms to solve problems of supporting foreign trade and investment processes in countries subjected to restrictions on a wide range of its interactions with foreign countries, companies and international markets. Empirical results show that gold-backed stablecoins, being effective at hedging assets in certain situations, provide countries with the opportunity to distance themselves from traditional financial institutions and reserve currencies in the context of external operations. Digital trading and investment platforms created on its basis do not exclude the risks inherent to the instrument. Moreover, they are exacerbated by continuing and increasing sanctions pressure on the economy integrated with such platforms. However, at the same time, these assets remain one of the most effective ways to support foreign trade and investment processes in these countries. The thesis is proven using an informalized method based on expert evaluations regarding the possibility of digital platforms overcoming trade and investment sanctions, the effects of which on the Russian economy cannot yet be accurately predicted. The study proposes two scenarios for the development of these platforms, potentially expanding the boundaries of foreign trade and investment interaction of the country subjected to sanctions with international markets.

**Keywords:** stablecoins; risks in business economics and finance; digital trading and investment platforms; economic sanctions; digital currencies

## 1. Introduction

The conditions associated with the introduction of foreign trade and financial restrictions in the relations of individual countries forces them to create digital trade and investment platforms to overcome restrictions. This allows nations to support domestic production and demand and minimize the consequences of breaking foreign trade chains. Until recently, the creation of such platforms was based solely on economic and local considerations, not provoked by political factors. At the same time, the possibilities of such platforms solving general economic problems constitute a legitimate research gap, primarily because of the relative short history of large-scale sanction policies implemented in the context of the new financial alternative, born in the depths of cryptoeconomics.

The formation of digital trading and investment platforms implies the issuance of stablecoins, pegged to either fiat currency or precious metals, for the purpose of settling foreign trade obligations. The second option of provision seems more appropriate for the purpose of reviving foreign trade and creating a parallel settlement system functioning in the decentralized financial sphere, considering that foreign trade restrictions are most often accompanied by financial sanctions.

The purpose of this paper is to analyze precious metal-backed stablecoins as a tool to overcome a wide range of sanctions restrictions. Research in this area is relevant both theoretically and practically, as it can expand the understanding of stablecoins' functionality, as well as provide new opportunities for international settlements of countries both subjected

to financial and trade sanctions and states with limited sovereignty of their national monetary systems. For companies under restrictive conditions, the inclusion of digital trading platforms created on the basis of stablecoin in operations solves the problem of, if not preservation, then maintenance at a minimum level of foreign trade contacts. Obviously, the solution to this problem can make a certain contribution to the stabilization of the macroeconomic situation, as well as, be subject to the mitigation and removal of sanctions from the strategic perspective to help bring the economy to the trajectory of sustainable development. The problem of ensuring this sustainability not only does not disappear under the conditions of sanctions, but it only becomes more acute. The provisions related to the substantiation of possible solutions to minimize the consequences of the restrictions imposed on the national economy through the creation of trade and investment platforms functioning on the basis of stablecoins, the efficiency of measures to include stablecoins in the contours of the traditional financial system in order to ensure the sustainability of the system's operations and the entire economy is the main content of the research undertaken in the framework of this work.

## 2. Literature Review

The ideas of justice and fairness in settlements in the interests of all their participants are associated with the search for stable instruments for their conduct. It can be said that they constitute one of the most important directions in the development of monetary theory, especially during the period of intensified interstate trade cooperation. The practical embodiment of these ideas goes back to the era of initial capital accumulation. Already at this stage, the similarity of the operations of early European deposit banks with today's supply of stablecoins is noted (Frost et al. 2020; Knot 2019). The technological and financial innovations in the latter half of the 20th century related to the expansion of the use of bank cards in settlements not least relied on the idea of value preservation (Arner et al. 2020). This and subsequent historical periods provide proof of the active response of the financial system to the interests of business development and the needs of citizens. These periods also coincide with growing dissatisfaction with the financial hegemony of developed countries, which often depend on national monetary systems to conduct cross-border payments and attack the safety of their reserves. The emergence and development of distributed ledger technology has created digital money, thereby broadening the understanding of monetary reality. This reality is now only a symbol of the challenge to systems based on fiat money, but it is already strengthening its role as an alternative to these systems. The phenomenon of stablecoin created on its foundation serves today as a means of settling foreign trade obligations and automated financial products, catalyzing and strengthening the cryptoindustry. This function provides prerequisites for the expansion of the boundaries of the analyzed phenomenon and the emergence and development of competition between different types of money, different in terms of the subjectivity of their emergence. The benefits of such competition, appearing on the basis of denationalization of money, were convincingly presented in his time by F. Hayek (1976). He argued the possibility of productive circulation of different currencies, refuting the thesis still expressed by W. Jevons that "there is nothing less suitable for competition than money" (Jevons 1875). By doing this, the author also contrasted his position with the "dogmas" of quantitative theory, a universalism whose inviolability is still questioned by many today (Friedman and Schwartz 1970).

Addressing the phenomenon of private money, many confirmations of the issue and circulation of non-public means of payment can be found in economic history. The culminating phase of a process of development of the phenomenon of private money broken in time is the denationalization of money, now accelerated by the technological factor. In this phase, as well as at the time of the pioneering concept of F. Hayek, the question of means of payment of private origin if often bought led to the denial of their monetary status. Such a discourse seems unproductive because it actually excludes an important aspect of modern monetary reality from the analysis.

This study examines the phenomenon of the decentralized issuance of stablecoins under the circumstances of government support in order to overcome trade and financial constraints external to the country. This event has hardly been studied in this perspective. As a result, this section will be limited to considering the main positions on the essence of stablecoins, as well as the peculiarities of its use and regulation.

The essence of stablecoin is quite clearly expressed in the definitions contained in the modern analysis of this phenomenon (Bullmann et al. 2019; Lipton et al. 2020; Sameeh 2018; Samman and Masanto 2019). Their advantages from other interpretations are that they emphasize the basic, conceptual elements of stablecoins, rather than the details of its implementation. This view is technologically neutral. Excluding existing forms of currency, it focuses on the ability of this instrument to mitigate the high volatility inherent to cryptocurrencies and, as a result, the potential for use in international transactions. The increase in the number and volume of the latter is associated with three circumstances: their relative speed and low-cost, the possibility of reaching the target audience with no or limited access to banking services, and the benefits for businesses and citizens of countries with unstable currencies (Bolliger 2019). These characteristics essentially describe the functional components of the instrument, highlighting the contours of the basic conditions, area of origin and nature of factors for its use for settlements and investment purposes. In terms of the ability to implement the latter in relation to stablecoins, it is important to keep in mind that they have been proven to be an elegant solution for developing the entire ecosystem of cryptocurrency trading while minimizing dependence on traditional banking services (Bolliger 2019).

Based on the wide array of stablecoins, the classification of which is mostly based on the type of collateral, the possibility of centralization, etc., it is necessary to emphasize the features of those which are tied to gold (Bolliger 2019; Dell'Erba 2019). These peculiarities have a definite influence on the investment qualities of stablecoins tied to metal. Among these characteristics of positive properties are stability, ease of understanding and perception and trust, including that provided by the low probability of cyberattacks. The flipside of these obvious advantages are no less obvious flaws: centralization and the associated dependence on the issuer, the need for frequent and independent auditing, dependence on the price dynamics of gold, the possibility of fictitious or incomplete security and potential problems with liquidity. Moreover, stablecoins, secured by sufficiently reliable and liquid collateral, can potentially serve as a digital currency-shelter in periods of cryptocurrency market crises. The phenomenon of stablecoins has been investigated in detail in this regard, which is adjacent to the aspects that the current study aims to examine (Liao and Caramichael 2022; Malloy and Lowe 2021; Wang et al. 2020). Yet with different methods of posting collateral, stablecoins are not deprived of volatility, which has been highlighted by researchers (Chohan 2019). Authors note a reduction in the probability of excess volatility if the supply of stablecoins is fully covered by fiat currency and guarantees of a third party acting as a pledge trustee (Jarno and Kołodziejczyk 2021). Collateralization of stablecoins with gold also does not guarantee the desired stability. It is dependent on fluctuations in the price of this metal. Moreover, geopolitical risks have been shown to increase the vulnerability of gold-backed stablecoins (Aloui et al. 2021).

Considering all these risks, digital platforms based on the issuance of stablecoins can be created for the purposes of settlements by business structures of countries affected by foreign trade and financial restrictions. These motives of regulatory arbitrage have been embodied in solutions for a number of years and have been investigated in sufficient detail. Simultaneously, the possibilities of combining the settlement and investment functionality of these platforms constitutes the content of the new challenge, especially in relation to the conditions and development tasks of countries under trade and financial sanctions.

The investment appeal of the proposed projects depends on the scale and potential sources of income of the issuer of stablecoins. The composition of revenue varies significantly depending on the specific token issuance scheme. Five revenue streams are usually distinguished:

- interest income, which, depending on the size of the reserve funds, can vary. Generally, issuers have an incentive to issue stablecoins for currencies that offer positive interest rates. For example, TrustToken maintains stablecoins for USD, GBP, AUD, CAD and HKD, not all of which have provided positive interest rates in the past (TrustToken 2020);
- transaction fees, which can be seen as a "last resort" (Tether) when the interest income outweighs the benefits of their introduction (Etherscan 2020);
- issuance and redemption fees. Stablecoin issuers can charge fees for their issuance (mining) and redemption (Tether 2020);
- cross-selling. Token issuers can cross-sell additional services that are based on their stablecoins. For example, some cryptocurrency exchanges are closely related to stable-coin issuers (e.g., Bitfinex and Tether). Stablecoins can serve as a means to attract and facilitate trading on their platforms;
- secondary tokens designed to increase in value as the stablecoin is used. System initiators regularly allocate a portion of these tokens to themselves to benefit from their increased value. For example, DAI has a special management token (MKR), which is also needed to close the Collateral Debt Position.

These revenue streams are generalized regardless of the type of stablecoin binding and the scale of its centralization. However, in one way or another, they can be extended to gold-backed tokens, and their utilization should be seen as an additional incentive factor in addition to the benefits provided by the DeFi sphere. In addition, it is inevitable that established stablecoin projects must take into account regulatory requirements and recommendations. This is an important aspect of the studied problem, which is covered in detail in the current analysis. Therefore, the study is limited to a review of the framework regulatory conditions accompanying the creation and functioning of stablecoin projects.

The document developed by the relevant ministers and governors of the G7 countries is the most important legal paper regarding the focus and content of national regulatory practices with respect to stablecoins (G7 Working Group on Stablecoins 2019). The document is focused on existing platforms with a significant customer base, initiating projects to create a digital currency with collateral. The content of the recommendations presented in this document is related to the possible risks of transactions with stablecoins, regardless of the scale of these operations. It is about the need to monitor threats associated with illegal actions, cyberattacks, data leakage, legal uncertainty, etc. In connection with the latter, the document contains recommendations related to the requirement for a clear legal description of the created project and the regulation of initiatives implemented in the interjurisdictional field. In addition, the G7 recommendations call not to discriminate against the created projects, so that adopted regulations remain technology-neutral and do not hinder innovation, whilst also guaranteeing the safety and reliability of the platforms created, full information to potential investors and protection of their rights. Regardless of the purpose of establishing platforms that issue and operate stablecoins, national regulators will inevitably be concerned with meeting these recommendations. This necessity is conditioned by circumstances that determine both the specifics of the DeFi sphere within which stablecoins circulate and the essence of this instrument itself, which is not dwelled on in detail due to their detailed coverage in the framework of contemporary analysis of the phenomenon of cryptoeconomics.

The justification of the possibilities of creating digital platforms, involving the issue of stablecoins in settlement and investment is advisable to be preceded by the analysis of the consolidated positions of this entire market.

## 3. Methodology

The analysis is based on the following methodological assumptions. Digital currencies, by facilitating the transfer of value between counterparties, are now the basis for the creation of digital platforms that transcend national borders. The scale and network effects of their activities are difficult to predict, since they depend on the motivations of the potential participants of these platforms, which can change under the influence of both

regulatory and political practices. Due to the difficulty of formalizing these factors, the study focuses on exploring the nature and causes of this motivation, providing a rationale for the possibilities of creating digital platforms based on stablecoins themselves, their functionality and the main limitations in their activities aimed at overcoming regulatory and other barriers imposed on certain jurisdictions.

The goal of the research implies the need to analyze aggregate indicators of the cryptocurrency market in terms of the functioning of the stablecoins market segment, as well as indicators characterizing the scale of ICOs, because the current study initially proceeded from the need to combine the settlement and investment functions of the created digital platforms. This analysis requires reference to sites providing extensive financial information on a country's macroeconomic and political parameters and the trends of the stablecoin market, as well as those specializing on listing existing ICOs and providing information about them. The final stage of the analysis is to highlight the case of opportunities to restore and reinvigorate international trade in commodities and financial assets with the participation of business entities that are now artificially cut off from these processes.

## 4. Results

As illustrated by conducted analysis, Tether continues to dominate the stablecoin market. In general, stablecoins tied to fiat reserve currencies occupy the vast majority of this market. When Tether tokens first began trading on Bitfinex in 2015, their turnover was quite small. However, as the cryptoeconomy evolved, so did Tether's stablecoin. The latter allowed bypassing traditional wire transfers by providing an alternative payment mechanism, including one between exchanges, without being exposed to the volatility of cryptocurrency prices. After the 2018 cryptocurrency crash, it was suggested that Tether was used to inflate and manipulate Bitcoin prices (Upson 2020). Moreover, it was assumed that cryptocurrency exchanges actively encouraged the use of stablecoins to increase trading volumes, as they provided an opportunity for trading venues to be less dependent on unstable banking relationships (Griffin and Shams 2019). The latter circumstance, in conjunction with the transactional potential, essentially caused the growth of this segment of the crypto market, expressed in the indicators of the supply of stablecoins, which exceeded $165 billion by the end of 2021.

At the same time, the probability of deviations of stablecoin from the USD parity can be traced in overall trends, which somewhat undermines the confidence in the already established view of their stability. The volatility of such deviations can be different and depend on many factors, including market manipulations, as the Terra USD situation has shown. Some researchers also point to the possibility of the loss of stability of this instrument, regardless of its design (Chohan 2019; Jarno and Kołodziejczyk 2021).

At the end of 2021 and the beginning of 2022, there was a revival of this segment as a whole and its part, with the emergence of gold-linked stablecoins. This is largely due to the unprecedented depreciation of the world's main reserve currency—the U.S. Dollar, which, as speculated, could be overcome in the foreseeable future. This circumstance provoked the growth of use of tokens tied to gold, which outstripped the dynamics of the entire cryptomarket. Gold outperformed bitcoin in its growth, but the market value of all gold tokens is still three orders of magnitude lower than the total market value of bitcoins. At the same time, experts note the continued weak appeal of gold tokens for investors due to the rather sluggish dynamics of the metal price and its frequent downward correction. This is the main difference between the analyzed instrument and other ones, rotating in the sphere of centralized issuance management and circulation of stablecoins—tokens pegged to fiat currencies. The universal nature of the latter, inherent to money, provides an opportunity to invest in conservative instruments offered by DeFi. Gold, despite preserving the property of "former money" and its worthy place among authoritative assets, still excludes such an opportunity.

The factor restraining investments in gold tokens is the opinion of potential users that the correction is not always quick, often taking days or even weeks. In addition, a

gold-backed stablecoin is far from always confirming its position as a stable asset. Gold, at least according to its price dynamics in the last two decades, is not a stable asset in itself. It is also volatile, and its liquidity is not high. Keeping large stocks of gold is costly. In the circumstances of stability, not to mention negative momentum of indices, this fact plays against investors.

A large portion of gold-linked tokens seek to become analogous to gold exchange-traded funds. The latter buy the physical metal, sharing ownership of it through shares. However, gold or metal stackable tokens are inferior to the mentioned instrument in terms of regulation and reliability. In addition, they encapsulate the risks of private key loss, cyberattacks, regulatory uncertainty and insufficient liquidity. Obviously, these circumstances will be deterrents in building the investment potential of the respective projects.

Reducing the potential for these disincentives is achieved by diversifying the instruments, leveraging other crypto-assets that can be raised to pay investment fees. This provides an opportunity to expand the issuer's instruments, in particular to provide loans in cryptocurrency and other reputable stablecoins, as was done in 2019 by Paxos, which launched the gold-linked stablecoin Pax Gold. This further enabled emerging startups to offer PAXG-backed loans in both fiat currencies and PAX, TrueUSD and USDC stablecoins (Stablecoins 2020).

The results of the analysis allows for identifying two main conditions contributing to the potential of investing in stablecoin:

- the ability to conduct transactions in the presence and increasing constraints on a seamless, technologically and price-optimal basis;
- support for the entire ecosystem of cryptocurrency trading, assuming and actually accompanying the circumvention of the contours of the traditional banking system (TradFi).

The second circumstance, in addition to the first one, reflecting the main functionality of the created stablecoin projects, is important for evaluating the potential of increasing the supply of this instrument in the crypto market. The factors of such growth can be the properties immanent to stablecoin, describing its functionality of servicing the turnover of other crypto-assets.

It is these features that constitute the main factor for the creation of trading and investment platforms primarily for the purposes of regulatory arbitrage. At the same time, the practice of creating such platforms to circumvent foreign trade and financial restrictions is insignificant both in terms of time and scale. Some countries whose economies and financial systems have been under restrictions for some time—North Korea, Venezuela, Iran, Belarus—have resorted to it. Yet even with this short history, it is already possible to note the fragmented filling of the deficit of goods and technologies in connection with the use of the possibilities of the DeFi sphere.

Experts say that the main constraints to the implementation of these trade and investment platforms, which operate with crypto assets, are the fear of potential participants of secondary sanctions against them, the scale of the economies of these countries and, accordingly, the extent of their involvement in the system of global economic relations. How will this potential manifest itself in an economy that is sufficiently large and diversified, with a significant number and scale of technological links with the outside world? Will the factor of scale play a decisive role here, if not in full, then in partial and significant for the implementation of the goal of sustainable economic development, to compensate the deficit of financial resources, imported goods and technologies? Obviously, formalized methods of forecasting the effects of the functioning of digital trading and investment platforms that involve the crypto economy's capabilities are limited in their capabilities, due to the weakness and insufficiency of the statistical base to clarify the aforementioned possibility. Therefore, to clarify this possibility in the new reality for the Russian economy, an expert approach will be further used based on estimates of limitations in the implementation of platform solutions and the probability of changes in the sanctions regime, which depends mainly on factors non-economic in nature.

## 5. Discussion

Digital trade and investment platforms based on cryptocurrencies and stablecoins partly contribute to the goal of mitigating individual foreign trade and financial restrictions imposed by sanctions on individual countries. This possibility is recognized by financial authorities of the Russian Federation. Following the adoption of the law on digital financial assets and digital currencies, which imposed rather strict restrictions on the operation of cryptocurrencies, introducing an easing of regulations in the national legislation in order to create opportunities to support foreign trade and overcome other restrictions associated with sanctions has been planned.

However, it is necessary to mention the difficulties of launching and the subsequent functioning of digital projects of trade and investment profiles in the current environment. Foreign trade and financial restrictions inevitably restrain the expansion of created projects. Overcoming them at the first stages of their development will not be facilitated by the rather narrow location of the concentration of production chains supported by the interested parties of the projects. These shortcomings may be later minimized by extending the trading functionality of the created platforms to the investment one (Lisin et al. 2021). In this case, the composition of participants is expanded by those who are interested in obtaining investment income, as well as by those entities that have a need to refinance their activities. This expansion involves the procedures of initial placement of ICO tokens secondary to stablecoins. The study identifies two scenarios for the development of digital platforms for the issuance of stablecoins.

The first one is limited to the placement of exclusively issuable stablecoins. It envisages the distribution of stablecoins among interested participants, who, in turn, will be limited only by the transactional functionality of these tokens to conduct foreign trade transactions. Acquisition of stable gold-linked coins by potential participants can be envisaged in fiat currencies and cryptocurrencies. In both cases, the value of the provided medium of payment is fixed in stablecoins, and the collateral plays its stabilizing role. This is important not only when paying for tokens with volatile, decentralized issued digital payment instruments, but also when presenting fiat money subject to inflationary pressures.

Storage of raised funds can be organized in different ways. Firstly, it can be done by the investor himself in the so-called "cold" wallet, when the private key is possessed by the investor himself. Secondly, the storage of "cold" wallets with third parties is the services of companies of the corresponding profile, providing secure storage services (custodial service). In essence, both options boil down to the phenomenon of escrow. In the first case, decentralized, and in the second, centralized (Bortnik et al. 2022). As a result, smart contracts can effectively serve as a certain form of escrow manager, since they will only execute predetermined instructions that cannot be subsequently changed. In general, regardless of the method of escrow service used, the success of a project is related to the presence or absence of that service. A similar mechanism may well be involved in commodity exchanges. Funds in escrow will be deposited in full or in part at the moment of fulfillment of obligations in terms of deliveries stipulated by foreign trade contracts. The functionality of escrow accounts in this case is limited to the service of facilitating the fulfillment of obligations of project participants. Regardless, it is reasonable to use this option for reasons of expanding the range of services provided by the digital platform.

The second scenario involves the implementation of an alternative ICO scheme. According to it, additional ICO tokens are issued by the companies involved in the implementation of the relevant project. These tokens are placed to interested investors in exchange for stablecoins. The companies-customers of the ICO, in turn, get the opportunity to refinance their activities in stablecoins. Thus, this scenario involves an additional number of investors interested in the project in terms of fixing the value of volatile crypto-assets and fiat currency and receiving more significant revenue streams distributed in favor of ICO customer companies, which can use not only stablecoins for the implementation of the transactional component of their business, but also fiat money.

This scenario does not exclude, but rather complements the platform's foreign trade functionality. It is associated with more intensive, large-scale and diverse financial flows, as well as with increased risks, the minimization or prevention of which can be ensured by hedging procedures. The second scenario, called the investment scenario with a certain degree of conditionality, provides an opportunity to attract external financing of companies—ICO customers. It also expands the regulatory framework applied to the platform's activity, not only in terms of expanding the set of instruments, but also in terms of regulation of crowdfunding operations (Kazakova et al. 2021). The diversity of income streams of the platform—commissions from ICO applicant companies, placement of stablecoins, settlements with them and other assets under trust management, ICO tokens, etc. gives a certain reserve for it in terms of establishing preferences for companies—project participants. This increases the possibility of refinancing their activities and may constitute a significant incentive for potential participants of the project. For investors entering the project, the subjects of the potential demand for ICO tokens, it is advisable to establish a commission-free regime, as it has been practiced by many similar projects. This aspect should be considered as important, given that the funds of these investors are, in fact, the main source of increasing the refinancing potential of the companies—participants of the stablecoin project.

Most of the existing digital platforms created to circumvent foreign trade and financial restrictions have trading functionality. The article proposes to supplement it with investment opportunities, which, taking into account both the features of stable provision of investment products issued on the platform and their potential profitability, should help to attract participants, both for purely investment purposes and trading. In addition, given the measure of involvement of the companies covered by the project in addressing the issues of the modern ESG agenda, this policy can act as an additional motivating factor to join the platform. A pronounced ESG orientation of the project participants' activities is important already because many experts consider the current trade and financial restrictions to be temporary. Obviously, they will be gradually mitigated and eliminated as the problems that provoked their introduction are solved. On the whole, DeFi platform solutions can encourage companies to become more actively involved in solving modern social and environmental problems and in achieving their sustainable development goals. Such targeting of DeFi solutions could guide and define the content of future research into the capabilities of emerging cryptocurrency and stablecoin-based trading and investment platforms.

When assessing the possibilities of digital platform solutions with trading and investment purposes in relation to the conditions and tasks of development of the Russian economy in the context of sanctions, the limitations of their potential results are noted. Of course, the results of these solutions will be more significant than in the countries against which sanctions were adopted earlier. This will be influenced by the scale of the economy and the geographical orientation of Russian exports. This will also be facilitated by the export specialization of the Russian economy, which is not limited exclusively to energy products, but also consists of other raw materials that are in demand from countries that have not joined the large-scale sanctions against the Russian Federation. This is the main feature that distinguishes Russian decisions in the analyzed sphere from the decisions of other countries under financial and economic restrictions. The common thing will be fragmentation in solving the problems of replenishment of the formed commodity and technological deficit. Furthermore, this limitation, which at times can be expanded by considering the competitive prices of Russian export positions and investment opportunities of platform products, will most often be supported by a set of non-economic factors and the increasing probability of sanctions for potential participants of created platforms.

## 6. Conclusions

The scientific novelty of the study consists in the substantiation of the possibilities of digital trading and investment platforms created on the basis of stablecoin to bypass the sanctions restrictions imposed on trade and financial transactions of certain countries.

Binding stablecoins to precious metals and avoiding the operation of reserve fiat currencies provide the necessary stability for the settlement of trade transactions and minimize the dependence on the regulatory and supervisory authorities of the countries that initiated the restrictions. This creates a trading platform for the resumption and expansion of export–import transactions. Whereas previously digital platform decisions were made primarily for the purpose of regulatory arbitrage, maximizing financial results, and these goals were essentially the only driver of the entire crypto economy, today these decisions are initiated to circumvent deliberately imposed large-scale sanctions of a trade and financial nature. These decisions, being provoked by a political factor that complements the economic grounds for the development of the cryptoindustry, are associated with a certain positive effect for the economies of the countries that have been subjected to sanctions. This is evidenced by the experience of creation and operation of platforms in a number of countries, which have been in actual economic isolation until now.

The possibility of expanding the activities of trade and investment platforms directly insignificantly correlates with the scale of the economy and the measure of its involvement in the system of international economic relations. The most significant factors that will influence the measure of participation of companies from different jurisdictions in the functioning of the digital trade and investment platform created for "anti-sanctions" purposes will be, firstly, their high interest in the commodity export positions of the country that has fallen under sanctions restrictions. Second, the functionality of the created platforms, and, thirdly, the probability of secondary sanctions to the participants of the platform interaction. Moreover, we consider the last factor to be more significant, determining the local nature and corresponding effect of the entire policy of circumventing trade and financial restrictions. The limited nature of the effect achieved does not cancel the expediency of the implementation of platform solutions based on stablecoin.

The fact that the participants of the relevant projects are given the opportunity not only to make payments in stable account units with guaranteed collateral, with no barriers to reverse conversion into national currencies and precious metal, but also to invest in derivative tokens with the use of their primary placement instrument, are considered to be the most important conditions for scaling these projects. The combination of trading and investment functions in the created digital platforms is especially crucial. This potentially expands the range of project participants, which will probably include not only subjects of international trade relations, but also investors interested in generating income from the issue of derivative tokens within the ICO, as well as participants who are short of resources to refinance their activities.

A look at the trends and scope of the expansion of the crypto economy and the sphere of decentralized finance shows a pronounced bias in sentiment and a desire to challenge the financial hegemony of individual states. Both economic and political reasons give rise to this challenge. At the same time, their current prevailing opportunistic bases are eventually transformed into strategic attitudes related to the need to achieve sustainable development goals, to strengthen the responsibility of business and to turn to the numerous social and environmental problems of modern reality on a large scale.

**Funding:** This research received no external funding.

**Institutional Review Board Statement:** Not applicable.

**Informed Consent Statement:** Informed consent was obtained from all subjects involved in the study.

**Conflicts of Interest:** The authors declare no conflict of interest.

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
