# Peer review of "Stablecoin-Based Digital Trading and Investment Platforms and Their Potential in Overcoming Sanctions Restrictions"

_economies, doi:10.3390/economies10100246_

Round 1

Reviewer 1 Report (Previous Reviewer 2)

Thank You for improving Your manuscript.

I'm satisfied by corrections.

So, I accept the manuscript.

Author Response

We thank the reviewers and the editorial board for their evaluation of our work, their attention to it, and their time and effort. We would like to note that the article has been further improved in accordance to the comments of other lead editor. 

Reviewer 2 Report (Previous Reviewer 1)

The quality of the paper has been significantly improved.

Author Response

We thank the reviewers and the editorial board for their evaluation of our work, their attention to it, and their time and effort. We would like to note that the article has been further improved in accordance to the comments of other lead editor. 

This manuscript is a resubmission of an earlier submission. The following is a list of the peer review reports and author responses from that submission.

Round 1

Reviewer 1 Report

The topic of stablecoins as alternative to financial restrictions is an interesting path of research. To make your paper marketable, consider include a methodological/theoretical framework, then carry out a proper analysis.

Reviewer 2 Report

Dear Author, thank You for so interesting research.

  1. General concept comments.

The article is written on the relevant topic and is logically proved.

However, I'd recommend making some improvements to the structure of the article:

  1. Please kindly shorten section 1 Introduction (setting aside the description of the common questions) and describe the research gap. The relevance of the topic is written sufficiently well, but the description of the rest three issues could be essentially improved. Please kindly help the reader understand the following important issues: the research gap, the research question, and the aim of the research. To my mind, the research aim is wider than the reader could understand from the last paragraph of the Introduction "Considering stablecoins as a possible tool to overcome limitations not of local nature is a fairly new task.". I'd recommend developing the authors' idea by making the logical connection to the concept of sustainable development and the following purpose of the research: "Considering possible solutions for sustainable development of the financial system on the basis of stablecoins as a digital tool within a wide range of external limitations". The authors' contribution could be considered as the development of a theoretical approach concerning the assessment of achieving sustainable development goals in social and governing aspects based on the proposed stablecoin digital trading and investment platforms. I suggest starting the aim of the research from the theoretical concept supporting the logical connection with such ESG goals, as social and governing. 
  2. The second section 2. Literature Review could include a table showing the differences between the definitions and concepts of stablecoin-based digital platforms being described in considered sources. Please kindly support the logic of the following sentence in the last paragraph: "The aforementioned and other provisions form the basis for national and supranational practices of stablecoin issuance".
  3. The sentence at the beginning of the third section Results "The justification for creating digital platforms that involve the issuance of stablecoins for settlement and investment purposes is preceded by an analysis of the aggregate positions of this entire market." could be moved to the end of the second section. I'd recommend the authors consider the proposed features of the suggested digital platforms on the basis of the analysis.
  4. The fourth Discussion section could include the debate on the advantages of the proposed approach and the difficulties of implementing stablecoin-based digital platforms. Please kindly make this section separate from the Results section and add the topic for future research.
  5. Conclusion. All conclusions regarding all 5 previous sections. I suggest authors should start the conclusion from the words - The scientific novelty of the research... and so on. 

Please check the grammatical errors in the title of Figure 1 "Figure 1. The overall supplt of stablecoins", the word - supplt.

Could You please at the end of the resulting section clearly explain in some paragraphs how the researchers prove the propositions of digital platforms because of avoiding methodological inaccuracies.

  1. Scientific Novelty.

Please kindly describe the scientific contribution to the theory. For example, the researchers could consider defining the conceptual framework of theoretical fundamentals for the solution of making the financial system sustainable problem by developing the digital stablecoin digital platforms' approach allowing to achieve ESG goals (social and governing) newly explained in the article. The authors could present the role of the developed approach in achieving sustainable development goals adding logic to the theoretical novelty.

 I'd suggest the authors could consider the new title of the Article: "Stablecoin-based digital platforms for achieving ESG-goals in external restrictions".

  1. General questions.

I think the researchers should carefully describe the Discussion section on the connection between the proposed approach and digitalization concepts. For this purpose, I'd recommend adding to the literature review some references regarding the wider understanding of the digital transformation concept as a basis for Sustainable development implementation:

Khalid, B., Naumova, E. Digital transformation SCM in view of Covid-19 from Thailand SMEs perspective (2021) Global Challenges of Digital Transformation of Markets, pp. 49-66. https://www.scopus.com/inward/record.uri?eid=2-s2.0-85116780657&partnerID=40&md5=ba31fd0ae1e1f4171e6c6c7e95801880

Barykin, S.Y., Kapustina, I.V., Sergeev, S.M., Kalinina, O.V., Vilken, V.V., de la Poza, E., Putikhin, Y.Y., Volkova, L.V. Developing the physical distribution digital twin model within the trade network (2021) Academy of Strategic Management Journal, 20 (SpecialIssue2), pp. 1-18. https://www.scopus.com/inward/record.uri?eid=2-s2.0-85106875305&partnerID=40&md5=db6f042b3d2623c43c8b21e13f470776